# Consumption and Digestion of Plastics by Greater Hive Moth Larvae

**DOI:** 10.3390/insects15090645

**Published:** 2024-08-28

**Authors:** Andrés Felipe Arias-González, Luis David Gómez-Méndez, Adriana Sáenz-Aponte

**Affiliations:** 1Laboratorio de Control Biológico, Grupo de Biología de Plantas y Sistemas Productivos, Departamento de Biología, Facultad de Ciencias, Pontificia Universidad Javeriana, Cra/# 43-82, Bogotá D.C. 110231, Colombia; ariasg_andres@javeriana.edu.co; 2Laboratorio de Microbiología Ambiental y de Suelos, Grupo de Biotecnología Ambiental e Industrial (GBAI), Departamento de Microbiología, Facultad de Ciencias, Pontificia Universidad Javeriana, Bogotá D.C. 110231, Colombia; luis.gomez@javeriana.edu.co

**Keywords:** *Galleria mellonella*, face masks, oxo-biodegradable plastic, stomodeum, fragmentation, mesenteron, excreta, ecdysteroids

## Abstract

**Simple Summary:**

This study tested the use of *Galleria mellonella* larvae as a biodegradation strategy for various types of plastics that were provided to larvae as their sole food source. The physical and enzymatic action of the larvae fragmented and consumed the plastics in various proportions, but none were digested. This was confirmed by the presence of microplastics in excreta and in larval digestive tracts, which affected their life cycle continuity by inducing the early formation of pupae and reducing the number of eggs produced.

**Abstract:**

The accumulation and unsustainable management of plastic waste generate environmental pollution that affects ecosystems, wildlife, and human health. We studied the possibility of using the consumption and digestion of oxo-biodegradable, compostable plastics and polypropylene from face masks by the fifth-instar larvae of *G. mellonella* as a strategy for the sustainable management of plastic waste. We used Fourier transform infrared spectrophotometry (FTIR) to determine the percentage of consumption and presence of microplastics in the digestive tract and excreta for 10 treatments evaluated for 135 h. The effects of plastics on the continuity of the life cycle of the greater hive moth were also determined. We established that the larvae fragmented and consumed 35.2 ± 23% of the plastics evaluated, with significant differences between treatments. Larvae were able to consume more of the intermediate layers of masks (86.31%) than the other plastics. However, none of the plastics were digested. Instead, microplastics accumulated in the excreta, resulting in nutritional deficits that affected the continuity of the life cycle, including the induction of the early formation of pupae after 24 h and a reduction in the number of eggs laid by the females.

## 1. Introduction

Plastics are synthetic organic polymers that are mostly obtained from crude oil, natural gas, and plant biomass. Composed of long chains of carbon monomers joined by covalent bonds, they are characterized by their stability, resistance, flexibility, hydrophobicity, malleability, and low price. Breaking these carbon monomer chains requires large amounts of energy, making them highly resistant to degradation [1,2]. A notable increase in the use of plastic products, especially single-use ones, was observed during the COVID-19 pandemic. Single-use plastic products include food containers; packaging; and personal protective equipment such as masks, gloves, and gowns. This increase has caused significant accumulations of plastic waste [3].

The accumulation and unsustainable management of plastic waste in landfills and natural environments damage terrestrial and marine ecosystems, while the accidental ingestion of plastic waste and/or entrapment in it affects the lives of associated fauna [1]. In addition, the prolonged exposure of these wastes to UV radiation, heat, humidity, friction, and chemical corrosion fragment the material and generate micro- and nano-plastics [4,5,6]. Toxic compounds such as bisphenols and phthalates that are present in plastics are then able to enter the food chain, where they eventually cause hormonal alterations, developmental problems, and chronic diseases [7].

Physical and chemical strategies have been used to mitigate the environmental impact of plastics. Among the former are mechanical fractionation, thermo-degradation, and photo-degradation, while the latter include oxidation, hydrolysis, and corrosion [4,5,6,7,8]. However, while these processes decompose plastic, they are not sustainable due to the copious amounts of energy they consume and the toxic gases they emit [5]. Biodegradation processes offer sustainable alternatives. They take advantage of the physical and enzymatic characteristics of various organisms, which enable them to decompose certain types of plastics into simpler compounds under specific conditions [9,10]. Recent studies have focused on the larvae of the greater hive moth, *Galleria mellonella* (Lepidoptera: Pyralidae) [11], together with their enzymes and associated microbiota, as a potential strategy for the biological degradation of plastics.

In 2017, Bombelli and collaborators exposed a low-density polyethylene (LDPE) bag to 100 g of *G. mellonella* larvae. After 40 min of exposure, they recorded gnawing and a total loss of 92 mg of LDPE. Analysis of the remains of the bag with FTIR identified carbonyls and ethylene glycol, simple compounds derived from the decomposition of LDPE [12]. Based on these results, various investigations were launched to find the mechanism by which *G. mellonella* consumes and decomposes LDPE. Studies that focus on the larval chewing of these materials stand out [13]. They have investigated the presence of oxidoreductases in the saliva of larvae [14], mapped their intestinal microbiota [15,16,17], and looked at their use to degrade other plastics such as high-density polyethylene (HDPE), expanded polystyrene (PSE), polypropylene (PP), and polyurethane (PU) [18,19].

Despite these studies, the potential of the larvae of the greater hive moth to consume and digest plastics with characteristics similar to those already studied, such as oxo-biodegradable, compostable plastics, and PP face masks, remains unknown. This study investigates that issue by evaluating the consumption and possible digestion of various types of plastics by the fifth-instar larvae of *G. mellonella* and the influence of this diet on the continuity of their life cycle.

## 2. Materials and Methods

### 2.1. Obtaining Biological Material and Plastic

Breeding of *G. mellonella* in the Biological Control Laboratory of the Pontificia Universidad Javeriana provided 1800 fifth-instar larvae, which were subsequently kept in darkness and without food at 26 °C for 24 h.

Six types of plastic were obtained from the thin film laboratory of the Pontificia Universidad Javeriana: a transparent low-density polyethylene bag; an oxo-biodegradable plastic food carrier (high-density polyethylene with pro-oxidant additives); fragmented oxo-biodegradable plastic exposed to environmental conditions (light, 26 °C, and aeration) for one year (high-density polyethylene with oxidized pro-oxidant additives); a green polyester bag; an expanded polystyrene food container; and a polypropylene face mask.

### 2.2. Percentage Consumed

To measure the percentage of material consumed, we produced experimental units composed of a 9 cm Petri dish + 36 cm^2^ square/0.14 ± 0.01 g of the type of diet to be supplied + 10 fifth instar larvae. The diets tested were as follows: a natural diet (T1), a square of absorbent paper + 2 g of honey + 2 g of beeswax; LDPE (T2); oxo-biodegradable plastic (T3); fragmented oxo-biodegradable plastic (T4); compostable plastic (T5); PSE (T6); face masks (T7); the inner layer of face masks (T8); the middle layer of face masks (T9); and the external layer of face masks (T10).

A completely randomized trial was designed. It consisted of ten treatments, each with three replications and three repetitions in time. After the incubation time, the final weight of the diets and the percentage consumed were measured for each treatment.

### 2.3. Digestion of Plastics by G. mellonella

After 135 h had passed, the larvae and pupae from each treatment were placed in an Erlenmeyer flask with 100 mL of 17.5% (*w*/*v*) ethanol for 15 min at 120 rpm. Subsequently, the solution was filtered through a 25 µm sieve, and the larvae were transferred to 100 mL of hydrogen peroxide (H_2_O_2_) at 30% (*w*/*v*). They were incubated at 60 °C and 80 rpm for 24 h to digest materials from their bodies. The solution was then filtered again, and the residues were deposited in 100 mL of 20% (*w*/*v*) sodium chloride (NaCl) for observation under a Carl Zeiss Microscopy GmbH stereomicroscope, Thermo Fisher Scientific, Waltham, Massachusetts, USA [20].

Excreta collected from each treatment were characterized morphologically and compositionally using a Carl Zeiss Microscopy GmbH stereomicroscope and a Shimadzu IRTracer-100 Shimadzu, Sao Paulo, Brasil infrared spectrophotometer, respectively. In addition, a compositional analysis of the plastics was performed before treatment; the same infrared spectrophotometer was used for this purpose.

### 2.4. Life Cycle Continuity

Ten experimental units per treatment were arranged as described above until the development of *G. mellonella* was completed. The time of pupa formation and the adults’ emergence was recorded. When all adults emerged, they were kept in darkness at 26 °C for 10 days to obtain egg clutches. After time had passed, the adults were stored in empty Petri dishes and sacrificed. The adult lengths were measured using ImageJ, (1.54f, 2023) and sex ratios were recorded.

Subsequently, the eggs from each treatment were collected with a brush, gently detached from each experimental unit, and placed in a plastic container for weighing to estimate their quantity (Equation (1)). This assay was performed in triplicate.
(1)$$eggs\;\left(UND\right)=x\;g\times \frac{1200\;eggs\;}{0.09\;g\;}$$
where *x* corresponds to the weight of eggs collected in grams.

### 2.5. Statistical Analysis

A one-way ANOVA test was performed with IBM^®^ SPSS Statistics 29.0, 2022, followed by a Duncan post hoc test (*p* < 0.05) to verify significant differences in the percentage of material consumed among the treatments evaluated, the number of eggs, and adult sizes. The other variables were analyzed using descriptive statistics.

## 3. Results

### 3.1. Percentage Consumed

After 135 h of exposure, the *G. mellonella* larvae had consumed 70.5 ± 0.5% of their natural diet (T1). Consumption began at the edges of the wax fragments, forming smaller pieces distributed throughout the box, with which they partially covered their bodies (Figure 1a).

There were significant differences in the percentages of the plastic-based diets consumed by the larvae [F = 127,143; df = 9; *p* ≤ 0.001] (Figure 2).

The larvae in T9 chewed and consumed the edges of the plastic, exceeding the consumption of T1 by 15.8% (Figure 2). In addition to this, there was random gnawing within the center of the material, as well as on the sides, which resulted in irregular holes of various sizes (Figure 1i). This consumption pattern was similar to those observed in the other treatments with compostable plastics, expanded polystyrene (EPS), and face masks (T5–T10) (Figure 1e–j).

Unlike T9, in the other treatments, the percentage of consumption decreased compared with T1, with the external layer of the mask (T10) being the lowest, with a percentage of 13.2 ± 0.45% (Figure 2). It is worth highlighting that the EPS (T6), being thicker than the other plastics, caused the larvae to not only form holes but also tunnels within the material, where they remained during the evaluated time. On the other hand, in T2 and T3, the larvae consumed only the edges of the material (Figure 1b,c), while in T4, consumption was greater since the plastic was fragmented into small pieces that were distributed throughout the box (Figure 1d).

### 3.2. Digestion of Plastics by G. mellonella

Thick, translucent cuticles that were intact and had well-defined shapes were evident in the T1 residue solution (Figure 3a). The solution also contained fragmented, irregularly shaped cuticles (Figure 3b); cuticle fragments; particulate material; and translucent, whitish, and beige fibers of various sizes (Figure 3c).

The differences and irregularities observed in T1 were not found in the other treatments. However, in T3, we observed thick, rough, opaque fragments with translucent orange colors. These characteristics are like those of the plastic that had adhered to the inner part of the cuticle (Figure 4a) and was found floating in the saline solution (Figure 4b). These fragments were probably microplastics.

The characterization analysis of the excreta in T1 showed a variety of sizes of elongated masses ranging from 0.6 to 2.1 mm. These masses were irregular, oily, and shiny and had rough textures with beige, yellow, and brown tones (Figure 5a). On the other hand, the FTIR reading showed transmittance minima between 450 and 1700 cm^−1^ and wavelengths from 2800 to 3300 cm^−1^ (Figure 5b).

In contrast, excreta from plastic-based diets consisted of amorphous masses of various sizes from 0.4 to 2.3 mm. They were dry, opaque, soft, and spongy and had rough textures with colors similar to the plastics used in each treatment (Figure 6). The excreta of T4 presented a granular texture with incrustations of white and blue microplastics (Figure 6d), but excreta observed in treatments corresponding to face masks (Figure 6g–j) consisted of thin, rolled, and elongated fibers with the same coloration as the plastic used.

The composition analysis found that the infrared spectra of the excreta of T1 differed significantly from the spectra of all the other treatments, showing minimum transmittance that did not exceed 80% at similar wave numbers, specifically between 400 to 900; 1350 to 1500; and 2850 to 2900 cm^−1^ (Figure 7). On the other hand, the infrared spectra of the plastics in the other treatments showed a similar trend to the spectra of the excreta, with minimal transmittance and higher intensity at the same wavelengths.

### 3.3. Life Cycle Continuity

*G. mellonella* larvae exposed to T1 began pupation after 78 h, while those exposed to plastic-based diets began the process within the first 24 h (Figure 8a). As for the adults, the T1 moths began to emerge 30 days later, but emergence occurred after 22 days in the treatments within which larvae were fed with plastic (Figure 8b). In the EPS (T6), life cycle monitoring could not be conducted because the pupae and adults remained inside the material throughout the evaluation period. However, after 37 days, in T6 the same adults were recovered as in T5 (compostable plastic).

Ten days after the adults emerged, we observed that they had begun to lay eggs throughout the boxes. However, the number of eggs laid differed significantly between treatments [F = 53,666; df = 9; *p* ≤ 0.001] (Figure 9). In T1, five females and four males emerged, and we obtained 185 ± 6 eggs, approximately 37 eggs per female. T3 and T10 had the same sex ratio as T1, but we obtained 51 ± 5 eggs (11 eggs/female) in T3 and 72 ± 11 (12 eggs/female) in T10. In T4 and T9, six females and four males emerged, and only 62 ± 6 eggs (11 eggs/female) were obtained in T4 and 103 ± 9 (24 eggs/female) in T9. The relationships between the other treatments and T1 all followed similar patterns (Figure 9), with the same trend of a lower number of eggs laid per female.

Finally, the average length of the females in T1 was 1.50 ± 0.08 cm, but the lengths of females in all the other treatments ranged from 1.2 to 1.5 cm. In other words, they were all significantly shorter than those in T1 [F = 4.417; df = 9; *p* = 0.003]. On the other hand, there were no significant differences [F = 2.122; df = 9; *p* = 0.077] in the lengths of males, whose range was 1 to 1.5 cm.

## 4. Discussion

The fifth-instar larvae of *G. mellonella* consumed the plastics to which they were exposed, regardless of whether their natural diet was pollen, wax, and/or bee honey [21,22]. When food is scarce, larvae tend to voraciously consume any object present and store the nutrients necessary to survive and continue their life cycles [23]. This behavior is associated with chemical compounds, such as long-chain hydrocarbons (C12–C18) and volatile compounds (aromas), that resemble their natural diet. These compounds generate a response in the peripheral gustatory chemoreceptors of the larvae, which leads them to associate them with a reward that is usually related to growth [24]. In other words, larvae establish an association between chemical components present in plastics and macromolecules that are important for their development, such as carbohydrates and fatty acids.

On the other hand, the ability to gnaw and consume part of the plastic begins in the stomodeum (anterior part of the larva), which includes the oral cavity, pharynx, esophagus, and proventriculus [22]. The mechanical actions of the jaw muscles in the oral cavity produce a process that tears and breaks the plastic. Subsequently, the fragments of plastic advance through the esophagus until they reach the proventriculus, a muscular organ with ridges and sclerotized spines. This organ finishes the grinding and scraping of the material [24]. Simultaneously, a biochemical process mediated by enzymes secreted by the salivary glands lubricates and depolymerizes complex compounds into monomers [22,24]. Arylphorin (alpha subunit) and hexamerin have been identified [14]. These enzymes can cause oxidizing activity in polyethylene films, which weaken carbon bonds and generate short chains of the same material, as well as alkoxy radicals (alcohols, ketones, carboxylic acids, aldehydes, esters, and lactones) [4]. Furthermore, it has been shown that saliva also contains phenoloxidases and hemocyanins, which regulate the oxidation of aromatic groups present in some plastics [14].

Consumption differences may be associated with differences in the characteristics of plastics, especially different densities and consistencies. A high density and rigid composition generate high stability [25]. This may hinder the physical and enzymatic mechanisms of depolymerization, thereby resulting in decreased consumption and injuries due to plastic incrustations in the digestive tract of the larva. Rather than being metabolized, these incrustations are excreted in the form of microplastics [26]. Furthermore, a woven consistency interferes with the physical consumption process and causes the material to become entangled in the structures of the stomodeum and, therefore, prevents it from reaching the mesenteron [13].

After this process of physical and enzymatic decomposition, the plastic reaches the mesenteron, an elongated sac composed of epithelial cells and covered with a permeable membrane of chitin and glycoproteins [22]. There, nutrients are digested and absorbed through the action of hydrolases, proteases, lipases, esterases, and the intestinal microbiota of *G. mellonella* [27]. Enzymes hydrolyze and break the bonds of the carbohydrates, honey proteins, and fatty acids contained in the wax [28,29]. However, these enzymes do not break the carbon–carbon bonds of plastics. Nevertheless, microorganisms present in the *G. mellonella* microbiota such as *Acinetobacter* spp., *Pseudomonas* spp., *Penicillium* spp., *Enterobacter* sp., and *Nocardia asteroides* have been shown to have the capacity to break down oxidized carbon–carbon bonds and transform them into esters, lactones, carboxylic acids, ketones, alcohols, and aldehydes [15,16]. In addition to this, *Bacillus spp.* are associated with the breakdown of LDPE, EPS, and polyurethane bonds, and *Aspergillus flavus* oxidizes HDPE through the synthesis of laccases (LMCO) [17].

In T2 (LDPE), the percentage of material consumed did not exceed 35%. This may be due to the structural organization of the plastic components (random branches), which are not very crystalline, giving them low density (0.94 g/cm^3^) and making them flexible. The flexibility of T2 means that gnawing frays the plastic, resulting in entropic structures that interfere with the stomodeum so that consumption decreases [13,25]. The results obtained were like those of Bombelli et al. (2017), who showed that 100 *G. mellonella* larvae only consumed 92 mg of a polyethylene bag during 12 h of exposure [12]. Although the plastic was consumed, it was probably not digested. This is evidenced by the infrared spectra of the excreta, which show peaks at wave numbers 2900, 1450, and 750 cm^−1^, which are characteristic of compositions corresponding to low-density polyethylene [25]. The composition of the plastic (LDPE) exhibits the same functional groups as those found in the excreta; however, the spectra differ, indicating that there has been a physical and oxidative transformation of the original material. Additional specific tests are needed to check digestion.

On the other hand, T3 and T4 plastics are HDPE with pro-oxidant additives that are not flexible [6]. They are hard, rigid, and crystalline, which makes chewing and enzymatic processing by the larvae difficult [26]. In T4, the plastic was broken into small pieces and distributed throughout the experimental unit. In T4, the plastic was broken into small pieces and distributed throughout the experimental unit. This can be attributed to the fact that the pro-oxidant additives in the plastic were oxidized by the conditions it was exposed to before the test, making the polymer chains weak and susceptible to breaking caused by any physical action [4,6]. In these cases, the larvae could more easily consume fragmented oxo-biodegradable plastic than whole oxo-biodegradable plastic. Regardless of consumption, the infrared spectra showed three transmittance minima that correspond to carbon–carbon bonds and compounds (1500 cm^−1^), C-H (2900–2850 cm^−1^), and CH2 (900–400 cm^−1^). These values are similar to those found in the infrared spectra of plastics before treatment and in HDPE analyzed by Velandia Cabra [25], indicating that the excreta are mostly composed of microplastics. It is important to highlight that complete oxo-biodegradable plastic (T3) was the only complete plastic found in the gastrointestinal contents of the larvae. Its composition and rigidity may cause adhesion, entanglement, or encrustation in epithelial cells of the proctodeum.

The larvae consumed the highest percentage of the material of the middle layer of the mask (T9) (86.3 ± 2.9%). This layer is composed of long PP chains with a regular distribution of methyl groups (isotactic PP), which makes it very dense (0.95 g/cm^3^) [25]. Nevertheless, the larvae easily consumed it because it is not crystalline; it is lighter, smoother, and less porous than the other layers; and it is the only layer approximately 100 µm thick [30]. All these characteristics favor the stomodeum’s mechanical action. In contrast, much smaller amounts of the inner and outer layers (T8 and T10) were consumed: 20.7 ± 1.4% of the inner layer and 13.2 ± 0.4% of the outer layer. These are also composed of PP, but this PP is woven and atactic (with randomly organized methyl groups). Its rough texture possibly causes difficulties in the mechanical process of the oral cavities of larvae. Although different percentages of the three diets were consumed, none of the plastics were digested by the larvae. This was confirmed by the infrared spectra of the excreta, which show that they are mostly composed of PP. The characterization of the material in the study by Velandia Cabra (2018) supports the results of this study. The infrared spectrum in that study was like our findings. Its local transmittance minima were at 2900 cm^−1^, 1350–1450 cm^−1^, and 1200–1000 cm^−1^, which correspond to compounds with C-H, C-C, and CH_3_ groups, respectively [25].

The plastic of T5 was polyester, while that of T6 was EPS. Both plastics are made of carbon chains with aromatic rings. The breakdown of these bonds is mediated by phenoloxidases, which oxidize the aromatic ring, weakening its bond with the carbon chains [14,31]. Studies in 2022 evaluated the consumption of PE, EPS, and PP by *G. mellonella* larvae and found that the larvae consumed a greater percentage of EPS than that of the other two plastics (57 mg) [19]. Those results are similar to this study’s results, in which the consumption of EPS was greater than the consumption of LDPE (T2) and PP from the internal and external layers of masks (T8–T10).

The *G. mellonella* larvae did not digest any of the plastics evaluated. In other words, the material was not metabolized. This agrees with studies that marked the carbon atoms of LDPE, in which it was observed that LDPE was excreted rather than being converted into insect biomass [26]. Furthermore, studies that performed compositional analyses of the excreta of larvae exposed to plastic diets have shown that the largest portions of the excreta have been short fragments of used polymers [13,15,18].

*G. mellonella* larvae have a relatively short life cycle of 45 days, but the only feeding stage is the larval stage. The larvae must consume enough to produce sufficient reserves of specific macronutrients and micronutrients for adequate development and subsequent reproduction during the adult stage [32]. Their natural diet consists mainly of pollen, wax, and honey [23]. Honey provides great nutritional value since it contains a high concentration of water, carbohydrates (fructose), vitamins, minerals, and fatty acids plus proteins in low concentration [28]. Wax is composed of long-chain fatty acids (>18C), hydrocarbons, alcohols, aldehydes, ketones, esters, sterols, triterpenes, and flavonoids [29]. Due to the nutritional contribution of these two substances, the larvae in T1 (natural diet) were able to molt the following larval instars, form the pupa, emerge as adults, and reproduce.

Plastic-based diets in these treatments were a source of carbon and energy. The depolymerization of the material only produces short chains of the material, alkoxy radicals [4], and compounds such as ethylene glycol and ketones of 10 to 18 carbons [12,14]. The lack of fatty acids, proteins, and micronutrients stimulated the early formation of prepupae and pupae and, in turn, the rapid emergence of adults. On the one hand, glycoproteins, phosphoproteins, sterols, and amino acids are necessary for the development of the organism because they fulfill essential functions for obtaining energy, growth, and cellular development [27,33]. Low concentrations or the absence of these compounds reduces the synthesis of body fat and molting hormones (ecdysteroids), and this interrupts ecdysis and causes the metamorphic cycle of the larva, inducing early pupa formation [33]. In other words, the organism uses the components mentioned above to transform itself into an adult rather than for the development of larval instars.

In addition to these effects, the reproductive system of future females is affected by the ingestion of plastic. Their reproductive systems consist of a pair of ovaries with 4 to 8 ovarioles. For the system to mature, a high cell load and the synthesis of juvenile hormones (JHs) are required. The latter induces the coupling of the ovarioles and the oviducts [24]. A large amount of protein is required to produce both the hormone and the required cell load. The absence of protein results in the immaturity of the reproductive cavities (a decrease in the number of ovarioles), which causes a reduction in the space where the eggs are produced, and this, in turn, affects the length of the female. Furthermore, egg formation begins in the oocyte, which is covered by a yolk (deutoplasm rich in lipids, polyunsaturated fatty acids, proteins, and carbohydrates) and a chorion (a layer of follicular cells that undergo apoptosis). These confer rigidity, hardness, and permeability. After this, the egg is fertilized, and the nutrients contained in the yolk are used for the formation of the embryo [24]. In our study, the low concentration of nutrients in diets with plastic interfered with the development of egg structures and embryo formation. This resulted in the production of fewer eggs with plastic-based diet treatments than with the natural diet.

These findings underline the limited capacity of *G. mellonella* larvae to metabolize plastics and the fact that these plastics affect their life cycles. Additional research is needed to develop a better understanding of the biological mechanisms involved. Other model organisms could be instrumental in developing future plastic degradation and bioconversion strategies, as suggested by recent studies highlighting the importance of the gut microbiome and enzymatic repertoire of insects such as *Tenebrio molitor* (Coleoptera: Tenebrionidae), and *Zophobas atratus* (Coleoptera: Tenebrionidae) in the degradation of plastic polymers such as LDPE, HDPE, PSE, and other plastics [18,34]. Also, the genetic manipulation capacity of organisms such as *Drosophila melanogaster* makes it highly desirable to combine plastic degradation potential with microbiome action, paving the way for new plastic waste management strategies [34]. In addition, our findings highlight the importance of addressing the problem of plastic pollution from various perspectives, including sustainable strategies for reducing the amount of these materials in the environment and minimizing their impacts on living organisms.

## Figures and Tables

**Figure 1 insects-15-00645-f001:**
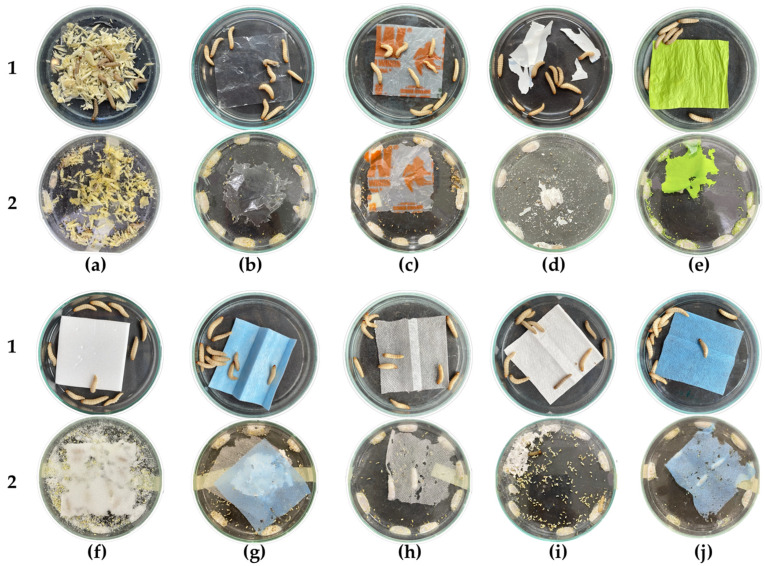
Exposure of *G. mellonella* larvae to different types of diet: (1) hour 0; (2) hour 135. (**a**) T1: Wax and honey. (**b**) T2: LDPE. (**c**) T3: Oxo-biodegradable plastic. (**d**) T4: Fragmented oxo-biodegradable plastic. (**e**) T5: Compostable plastic. (**f**) T6: PSE. (**g**) T7: PP face masks. (**h**) T8: PP inner layer of face masks. (**i**) T9: PP middle layer of face masks. (**j**) T10: PP outer layer of face masks.

**Figure 2 insects-15-00645-f002:**
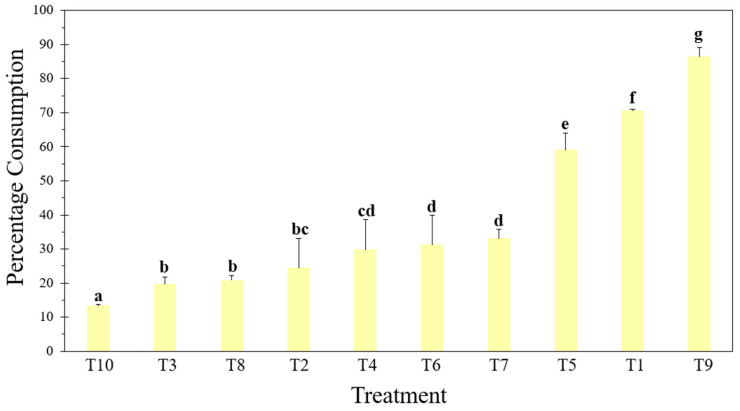
The percentage of material consumed by *G. mellonella* larvae exposed to different types of diet for 135 h. T1: Wax and honey. T2: LDPE. T3: Oxo-biodegradable plastic. T4: Fragmented oxo-biodegradable plastic. T5: Compostable plastic. T6: PSE. T7: PP face masks. T8: PP inner layer of face masks. T9: PP middle layer of face masks. T10: PP outer layer of face masks. Error bars represent the standard deviation of three replicates and three repetitions over time. Letters above the bars indicate significant differences (Duncan *p* ≤ 0.05).

**Figure 3 insects-15-00645-f003:**
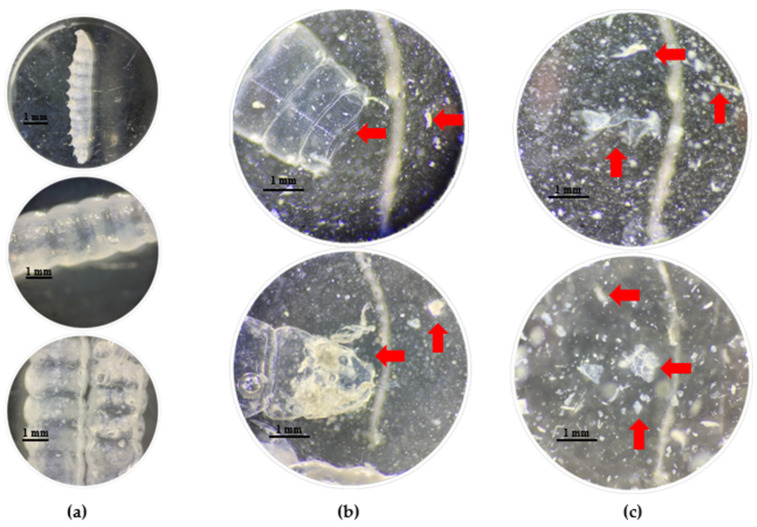
Residues of larvae in saline solution of T1 (wax and honey). (**a**) Intact cuticles. (**b**) Fragmented cuticles. (**c**) Cuticle fragments and wax. Arrows indicate cuticular residues.

**Figure 4 insects-15-00645-f004:**
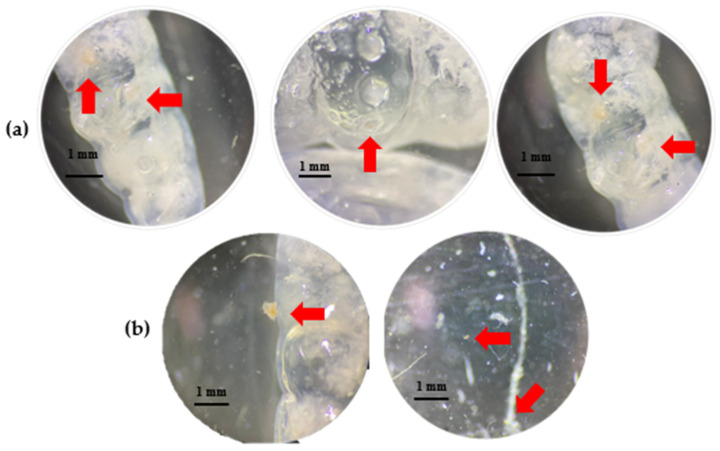
Residues of larvae in T3 (oxo-biodegradable plastic) saline solution. (**a**) Microplastics adhering to the cuticle of the larva. (**b**) Microplastics floating in the saline solution. Arrows indicate plastic waste.

**Figure 5 insects-15-00645-f005:**
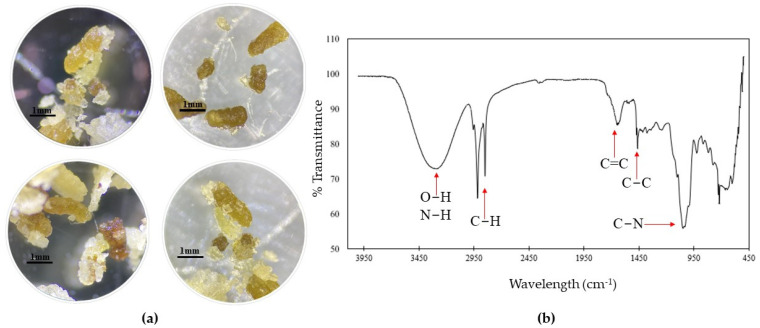
Characterization of *G. mellonella* excreta after consumption of bee wax and honey (T1). (**a**) T1 excreta. (**b**) FTIR of T1 excreta. The arrows indicate functional groups associated with local minima of transmittance at various wavelengths.

**Figure 6 insects-15-00645-f006:**
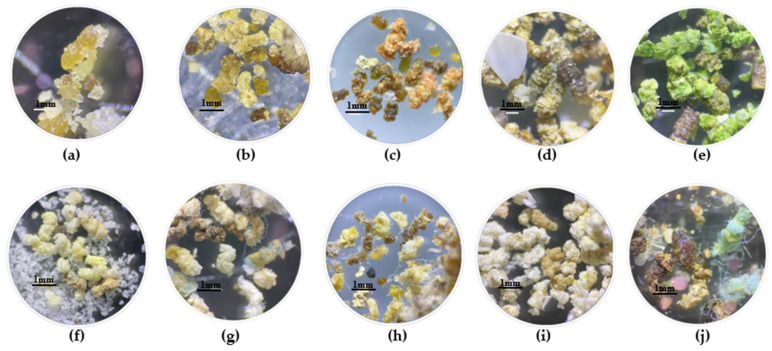
Microscopic photos of *G. mellonella* larval excreta from each treatment. (**a**) T1: Wax and honey. (**b**) T2: LDPE. (**c**) T3: Oxo-biodegradable plastic. (**d**) T4: Fragmented oxo-biodegradable plastic. (**e**) T5: Compostable plastic. (**f**) T6: PSE. (**g**) T7: PP face masks. (**h**) T8: PP inner layer of face masks. (**i**) T9: PP middle layer of face masks. (**j**) T10: PP outer layer of face masks.

**Figure 7 insects-15-00645-f007:**
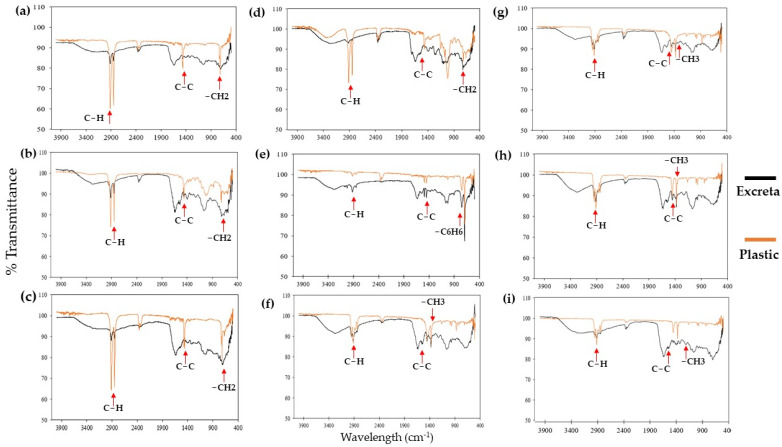
Infrared spectra of excreta and plastic (control) from treatments. (**a**) T2: LDPE. (**b**) T3: Oxo-biodegradable plastic. (**c**) T4: Fragmented oxo-biodegradable plastic. (**d**) T5: Compostable plastic. (**e**) T6: PSE. (**f**) T7: PP face masks. (**g**) T8: PP inner layer of face masks. (**h**) T9: PP middle layer of face masks. (**i**) T10: PP outer layer of face masks.

**Figure 8 insects-15-00645-f008:**
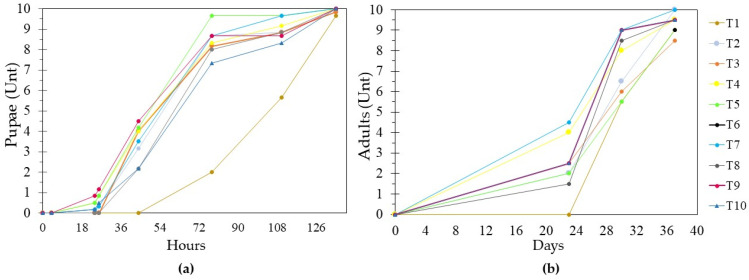
Continuity of the life cycle of *G. mellonella* after treatment. (**a**) Pupation. (**b**) Emergence of adults according to the diet provided. T1: Wax and honey. T2: LDPE. T3: Oxo-biodegradable plastic. T4: Fragmented oxo-biodegradable plastic. T5: Compostable plastic. T6: PSE. T7: PP face masks. T8: PP inner layer of face masks. T9: PP middle layer of face masks. T10: PP outer layer of face masks.

**Figure 9 insects-15-00645-f009:**
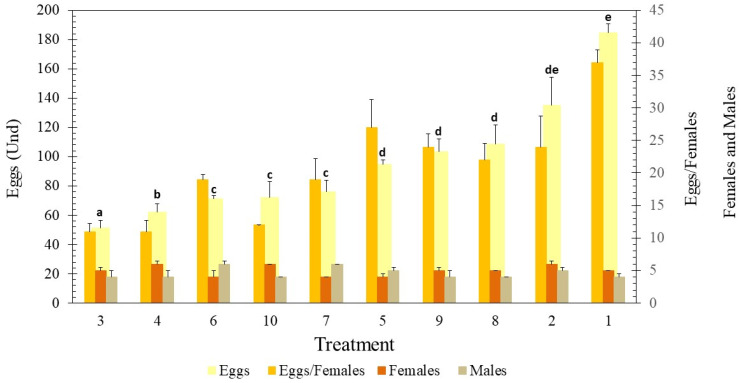
Number of eggs, eggs/females, and sex ratios of *G. mellonella* per treatment. T1: Wax and honey. T2: LDPE. T3: Oxo-biodegradable plastic. T4: Fragmented oxo-biodegradable plastic. T5: Compostable plastic. T6: PSE. T7: PP face masks. T8: PP inner layer of face masks. T9: PP middle layer of face masks. T10: PP outer layer of face masks. Error bars represent the standard deviation of two replicates and two repetitions over time. The letters in each column represent significant differences (Duncan *p* ≤ 0.05) in the number of eggs laid in each treatment.

## Data Availability

The dataset is available upon request from the authors.

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
