# Peer review of "Consumption and Digestion of Plastics by Greater Hive Moth Larvae"

_insects, 2024, doi:10.3390/insects15090645_

Round 1

Reviewer 1 Report

Comments and Suggestions for Authors

The manuscript deals with the use of insects as plastic biodegraders. The authors confirm that G. mellonella are voracious consumer of many type of polymers when other food sources are scarce. This was already shown in previous papers. The authors also show that plastic ingestion induces early pupation and impacts the life cycle of the moth.

Overall, the manuscript is clearly written. I have some comments (reported below) which will hopefully improve the manuscript. 

The type of material supplied in each treatment should be stated early in the manuscript (e..g. in the methods section) rather than in the discussion

Life cycle continuity. The plot can be improved. Can be the graph changed so that dots are connected by (colored) lines? also charms should be bigger in size.

It is not sufficiently clear how the egg counting experiment was conducted. I suppose that the adults (or pupae) were removed from the original petri dish (containing the treatment). Where do you le them mate? How were the eggs collected? For how long did you collect eggs?

The eggs counted should be normalized for the number of females in each test.

l270 "Although the plastic was consumed, it was not digested" How can you be sure that it wasn't digested at all? Since you haven't tested for the plastic bio-conversion this statement should be rectified.

l235 "Arylphorin (an alpha subunit)" remove "an"

l277. "... this plastic was exposed to environmental conditions (light, 26° C and aeration) for a year" This information must be give in the methods

In my opinion, the manuscript lack positive controls in FTIR analyses. The authors should explain why this controls were not included.

The authors might want to mention in the discussion that model organisms could be also helpful in developing degradation and/or bioconversion strategies in the future (doi: 10.1016/j.scitotenv.2024.169942)

Comments on the Quality of English Language

English language is fine, few sentences need revision.

Author Response

Comments 1: The type of material supplied in each treatment should be stated early in the manuscript (e..g. in the methods section) rather than in the discussion

Response 1: Thank you for pointing this out. We agree with this comment. This information was added to the methodology section. You can review it on page 2, line 80 (title) and line 85-90.

Comments 2: Life cycle continuity. The plot can be improved. Can be the graph changed so that dots are connected by (colored) lines? also charms should be bigger in size.

Response 2: Agree. We have modified Figure 8 according to your comment. This was included in the text and was also attached as a jpg image to the new submission.

Comments 3: It is not sufficiently clear how the egg counting experiment was conducted. I suppose that the adults (or pupae) were removed from the original petri dish (containing the treatment). Where do you let them mate? How were the eggs collected? For how long did you collect eggs?

Response 3: Agree. We have included more information about the egg counting experiment in the manuscript. The changes can be found on page 3, line 114-124.

Comments 4: The eggs counted should be normalized for the number of females in each test.

Response 4:

We agree with this comment. This information was added to the results section. You can review it on page 8, line 223 - 228.

Comments 5: l270 "Although the plastic was consumed, it was not digested" How can you be sure that it wasn't digested at all? Since you haven't tested for the plastic bio-conversion this statement should be rectified.

Response 5: Agree. We changed the statement to: "Although the plastic was consumed, it was probably not digested" page 10, line 298 and 299. We also complemented this idea in line 302: "however, more specific tests are needed to check digestion"

Comments 6: l235 "Arylphorin (an alpha subunit)" remove "an"

Response 6: We remove that. Page 9, line 263 and 264

Comments 7: l277. "... this plastic was exposed to environmental conditions (light, 26° C and aeration) for a year" This information must be given in the methods

Response 7: We agree with this comment. This information was added to the methodology section. You can review it on page 2, line 80 (title) and line 85-91. Also, it was removed from the discussion section. This change is seen on page 10, line 306-310

Comments 8: In my opinion, the manuscript lack positive controls in FTIR analyses. The authors should explain why this controls were not included.

Response 8: Positive controls were verified, but not included, because these were equal to the infrared spectra of the excreta with plastic diets. Those have been previously reported in articles. For this essay, we take those from: https://doi.org/10.21158/23823399.v5.n0.2017.2005

Comments 9: The authors might want to mention in the discussion that model organisms could be also helpful in developing degradation and/or bioconversion strategies in the future (doi: 10.1016/j.scitotenv.2024.169942)

Response 9: We agree with this comment. This information was added to the discussion section. You can review it on page 11 - 12, line 389 - 397.

Reviewer 2 Report

Comments and Suggestions for Authors

The paper "Consumption and digestion of plastics by greater hive moth larvae" notes that the consumption and digestion of plastics with similar characteristics to those already studied, such as oxo-biodegradable, compostable, or PP face masks, remain unknown. This study investigates this issue by evaluating the consumption and possible digestion of various types of plastics by fifth-instar larvae of G. mellonella and the influence of this diet on the continuity of their life cycle.

The results highlight the limited capacity of G. mellonella larvae to metabolize plastics and the fact that these plastics affect their life cycles. More research is needed to develop a better understanding of the biological mechanisms involved. Additionally, our findings underscore the importance of addressing the plastic pollution problem from various perspectives, including sustainable strategies to reduce the amount of these materials in the environment and minimize their impacts on living organisms.

To better evaluate this study, I conducted a bibliographic review of the results on plastic consumption and digestion by the wax moth. I must say I was quite impressed with this work. It had been a while since I had seen a study with such well-done research and discussion. Moreover, the results are very important for developing measures to reduce microplastics in our environment. The moth was unable to digest the plastics it consumed, and it is indeed important to understand the pollution COVID-19 has brought with sanitization to avoid contaminations.

I have only a few suggestions:

  • Line 112: Is the equation a standard equation? Can you provide a reference for it?
  • Presentation of tables and figures: Instead of numbers 1, 2, 3, 4, it would be much more visual to substitute with T1, T2, T3, etc.
  • In the title of Figures and Tables, indicate what T1…T10 means.
  • Paragraph in line 142.
  • Lines 355 and 356: "This resulted in the production of fewer eggs with plastic-based diet treatments than with the natural diet" - elaborate on this topic.

Author Response

Comments 1: Line 112: Is the equation a standard equation? Can you provide a reference for it?

Response 1: This equation was established from the different standardizations carried out on the breeding of Galleria mellonella, existing in the biological control laboratory, which allows better management of the clutches to be placed in each box with diet, for the new larvae.

Comments 2: Presentation of tables and figures: Instead of numbers 1, 2, 3, 4, it would be much more visual to substitute with T1, T2, T3, etc.

Response 2: Agree. We changed Figure 2 (page 4), and Figure 9 (page 8). This was included in the text and was attached as a jpg image to the new submission.

Comments 3: In the title of Figures and Tables, indicate what T1…T10 means.

Response 3: Agree. We added the information to the figure's titles.

·       Figure 2 page 5, line 151-154

·       Figure 3 page 5, line 172

·       Figure 4 page 6, line 180

·       Figure 6 page 7, lines 198-201

·       Figure 7 page 7, lines 206-209

·       Figure 8 page 8, lines 217-219

·       Figure 9 page 8, lines 232-235

Comments 4: Paragraph in line 142.

Response 4: Agree. We changed. Page 5.

Comments 5: Lines 355 and 356: "This resulted in the production of fewer eggs with plastic-based diet treatments than with the natural diet" - elaborate on this topic.

Response 5: We think that the topic is elaborated in paragraph 4, page 11, lines 350-359.  

“In addition to these effects, the reproductive system of future females is affected by ingestion of plastic. Their reproductive systems consist of a pair of ovaries with 4 to 8 ovarioles. For the system to mature, a high cell load and the synthesis of juvenile hormone (JH), are required. The latter induces the coupling of the ovarioles and the oviducts [24]. A large amount of protein is required for production of both the hormone and the required cell load. Absence of protein results in immaturity of the reproductive cavities (decrease in the number of ovarioles) which causes a reduction in the space where the eggs are produced, and this in turn affects the length of the female. Furthermore, egg formation begins in the oocyte which is covered by yolk (deutoplasm rich in lipids, polyunsaturated fatty acids, proteins and carbohydrates) and a chorion (a layer of follicular cells that undergo apoptosis). These confer rigidity, hardness and permeability. After this, the egg is fertilized, and the nutrients contained in the yolk are used for the formation of the embryo [24]. The low concentration of nutrients in diets with plastic interfered with the development of egg structures and embryo formation. This resulted in production of fewer eggs with plastic-based diet treatments than with the natural diet”

However, what else do you think we need to elaborate?

Round 2

Reviewer 1 Report

Comments and Suggestions for Authors

I would like to thank the Aauthors for addressing the issues raised in the previous review round.  However, I have still a couple of comments.

Instead of simply citing the normalized egg count in the text, it should be reported in figure 9 .

If the Authors have measured the spectra of the positive control sample, they should be shown, even if they are similar to that observed or plastic excreta.

Comments on the Quality of English Language

English language is fine, just check for typos

Author Response

Comments 1: Instead of simply citing the normalized egg count in the text, it should be reported in figure 9.

Response 1: Thank you for pointing this out. We agree with this comment. This information was added in Figure 9. This was attached as a jpg image to the new submission.

We also added information on Line 237 (Page 9) and corrected some data in the presentation of the figure, (Page 8) lines 228 – 232.

Comments 2: If the Authors have measured the spectra of the positive control sample, they should be shown, even if they are similar to that observed or plastic excreta.

Response 2: Agree. We added positive controls in the methodology, results and discussion section.

Page 3, line 112-114.

Page 7, line 207-209 and line 212.

Page 10, line 303 – 310.

Page 10 and 11, line 319 – 324.

We also modified de Figure 7. This was attached as a jpg image to the new submission.

Round 3

Reviewer 1 Report

Comments and Suggestions for Authors

Thank you for addressing my comments. 

Please check the text carefully for grammar errors and typos (I found some which are listed below).

keywords and l381 "ecdesteroids"

l43  "This increase caused significant....." change to "This increase has  caused significant..."

l130 ."consumed between among treatments" change to "consumed among the treatments"

Comments on the Quality of English Language

there are still some grammar error and typos in the text.

Author Response

Comments 1:
Please check the text carefully for grammar errors and typos (I found some which are 
listed below).
keywords and l381 "ecdesteroids"
l43 "This increase caused significant....." change to "This increase has caused 
significant..."
l130 ."consumed between among treatments" change to "consumed among the treatments"

Response 1: Thank you for pointing this out. We agree with the comment and have 
corrected each of the grammatical errors that were underlined in the document
Page 1, line 32 
Page 2, line 42 and 43 
Page 3, line 131
Page 12, line 38